# TorchXRayVision:
# A library of chest X-ray datasets and models

**Joseph Paul Cohen**[S,M,Q,A] **Joseph D. Viviano**[Q] **Paul Bertin**[M,Q] **Paul Morrison**[Q,F]
**Parsa Torabian**[W] **Matteo Guarrera**[B,E,P] **Matthew P Lungren**[S,A]
**Akshay Chaudhari**[S,A] **Rupert Brooks**[N] **Mohammad Hashir**[M,Q] **Hadrien Bertrand**[Q]

[S]*Stanford University* [M]*Université de Montréal* [Q]*Mila, Quebec AI Institute* [A]*Center for Artificial Intelligence in Medicine & Imaging (AIMI)* [N]*Nuance Communications* [F]*Fontbonne University* [W]*University of Waterloo* [B]*University of California, Berkeley* [P]*Politecnico di Torino* [E]*EURECOM*
*Correspondance: https://github.com/mlmed/torchxrayvision/issues*

**Editors:** Under Review for MIDL 2022

## Abstract

TorchXRayVision is an open source software library for working with chest X-ray datasets and deep learning models. It provides a common interface and common pre-processing chain for a wide set of publicly available chest X-ray datasets. In addition, a number of classification and representation learning models with different architectures, trained on different data combinations, are available through the library to serve as baselines or feature extractors.

**Keywords:** Chest X-ray, CXR, pre-trained models, datasets, representation learning, generalization, feature extraction, PyTorch

## 1. Introduction

Chest X-rays are the most common medical imaging test in the world and represent the bulk of medical computer vision publications and open medical imaging data in the deep learning community [UK NHS, 2019]. Yet despite the large number of datasets and publications, it can be difficult for researchers to properly compare previous work and to investigate generalization across different datasets. Even when data and code are available, small but important differences in dataset organization, processing, or training procedures can significantly impact the results. This makes establishing meaningful baselines a strenuous task for researchers. In addition, constantly re-implementing the same dataloaders is not the best use of time. There is a need for common software infrastructure.

TorchXRayVision[1] (XRV) was created to address this difficulty by establishing a reusable framework for reproducible research and consistent baseline experiments. A key design objective is to provide a clear interface and separation between datasets and models. The library provides a common interface to multiple available chest X-ray datasets, which can easily be swapped out during model training and evaluation. Common pre- and post-processing components are provided and the datasets are compatible with torchvision [Paszke et al., 2019] components for augmentation. We include pre-trained and easily downloadable models which can be used directly for baseline comparisons or to generate feature vectors for downstream tasks.

Three specific use cases where the project has already proved useful include:

---

1. https://github.com/mlmed/torchxrayvision

- **Evaluating models:** It is important to rigorously evaluate the robustness of models using multiple external datasets. However, associated clinical data with each dataset can vary greatly which makes it difficult to apply methods to multiple datasets. TorchXRayVision provides access to many datasets in a uniform way so that they can be swapped out with a single line of code. These datasets can also be merged and filtered to construct specific distributional shifts for studying generalization.

- **Developing models:** Making it easier for Deep Learning researchers to work on medical tasks. Pre-trained models are useful for baseline comparisons as well as feature extractors. These pre-trained models have already been used for transfer learning to related chest X-ray tasks such as patient severity scoring [Cohen et al., 2020a; Gomes et al., 2020a] and predicting aspects about a patients clinical trajectory [Cohen et al., 2020c; Gomes et al., 2020b; Maurya, 2020]. As illustrated in [Cherti and Jitsev, 2021] TorchXRayVision pre-trained models are used to study few-shot transfer learning. The pre-trained models are used as feature extractors of images for multi-modal models [Delbrouck et al., 2021; Soenksen et al., 2022]. In [Sundaram and Hulkund, 2021] the library is used for baseline models as well as to explore methods of generative adversarial network-based (GAN) data augmentation. The pre-trained models and training pipeline were used in [Tetteh et al., 2021] to study balancing batches while training on multiple datasets.

- **Studying model failures and limitations:** The many pre-trained models provided are not perfect and can be studied to determine how they fail, which can inform the development of better models. Also, the many datasets available in TorchXRayVision make it possible to study out-of-distribution generalization when covariate and concept shifts are present. The library was initially developed for this purpose in [Cohen et al., 2020b]. This library has already been used in work by [Robinson et al., 2021] which focused on studying shortcut learning caused by covariate shift in chest X-ray models. Work by [Viviano et al., 2020] explored failures in saliency maps using this library and special utilities are included to produce datasets with different types of covariate shift and spurious correlations. Work by [Cohen et al., 2021a] generated counterfactual explanations for model predictions using the classifiers and autoencoder in the library.

A key design consideration when developing this library was to follow an object-oriented approach. This turns the various components of an experiment into objects which can be easily swapped. The primary objects are datasets and models (including pre-trained weights). A suite of utilities for working with these objects are also included. The library is available in Python via pip with the package name `torchxrayvision` and is typically imported as `import torchxrayvision as xrv`. It is based on PyTorch [Paszke et al., 2019] and modeled after the torchvision library (hence the name). TorchXRayVision's compatibility and conformity to established convention makes adoption intuitive for practitioners.

## 2. Models

The library is composed of core and baseline classifiers. Core classifiers are trained specifically for this library (initially trained for [Cohen et al., 2020b]) and baseline classifiers come from other papers that have been adapted to provide the same interface and work with

the same input pixel scaling as our core models (see `BENCHMARKS.md` on GitHub for current model performances). All models automatically resize input images (higher or lower using bilinear interpolation) to match the specified size they were trained on. This allows them to be easily swapped out for experiments. See §3.1 for details on image pro-processing. Pre-trained models are hosted on GitHub and automatically downloaded to the user's local `~/.torchxrayvision` directory.

## 2.1. Core Classifiers

Core pre-trained classifiers are provided as PyTorch Modules which are fully differentiable in order to work seamlessly with other PyTorch code. Models are specified using the "weights" parameter which have the general form of:

$$\underbrace{\text{densenet121}}_{\text{Architecture}}\text{-}\overbrace{\text{res224-}}^{\text{Resolution}}\underbrace{\text{rsna}}_{\text{Training dataset}}$$

Each pre-trained model aims to have 18 independent output classes as defined in `xrv.datasets.default_pathologies`. However, since not all datasets provide each pathology class, some will return `NaN` as the prediction of that pathology. The models indicating "all" as the dataset have been trained on as many datasets were available at the time. More details about each set of weights is available at the head of the `models.py` file. The current core classifiers were trained with data augmentation to improve generalization. According to best data augmentation parameters found in Cohen et al. [2019], each image was randomly rotated up to 45 degrees, translated up to 15% and scaled larger or smaller up to 10%. These models are trained using binary cross entropy losses, Adam optimizer [Kingma and Ba, 2014], a batch size of 64, and a learning rate of 0.001.

```python
# DenseNet 224x224 model trained on multiple datasets
model = xrv.models.DenseNet(weights="densenet121-res224-all")
# DenseNet trained on just the RSNA Pneumonia dataset
model = xrv.models.DenseNet(weights="densenet121-res224-rsna")
# ResNet 512x512 model trained on multiple datasets
model = xrv.models.ResNet(weights="resnet50-res512-all")
```

## 2.2. Baseline Classifiers

Currently there are two baseline classifiers from other research groups which were added to make it easier to compare against TorchXRayVision core models. The models adhere to the same interface as the core models and can be easily swapped out. The first is a JFHealthcare model [Ye et al., 2020] trained on CheXpert data [Irvin et al., 2019] and the second is an official CheXpert model [Irvin et al., 2019] trained by the CheXpert team.

```python
# DenseNet121 from JF Healthcare for the CheXpert competition
model = xrv.baseline_models.jfhealthcare.DenseNet()
# Official Stanford CheXpert model
model = xrv.baseline_models.chexpert.DenseNet()
```

The weights for the CheXpert model are very (6GB) and have been hosted on Academic Torrents [Cohen and Lo, 2014] as well as the Internet Archive.

## 2.3. Classifier Interface

Each classifier provides a field `model.pathologies` which aligns to the list of predictions that the model makes. Depending on the weights loaded this list will change. The predictions can be aligned to pathology names as follows:

```
predictions = model(img)[0] # 0 is first element of batch
dict(zip(model.pathologies,predictions.detach().numpy()))
# output:
{'Atelectasis': 0.3566849,
 'Consolidation': 0.72457345,
 'Infiltration': 0.8974177, ...}
```

Getting a specific output can be achieved as follows. The outputs remain part of the computation graph and can therefore be embedded in a larger network.

```
prediction = model(img)[:,model.pathologies.index("Consolidation")]
```

## 2.4. Feature Extraction

The pre-trained models can also be used as features extractors for semi-supervised training or transfer learning tasks. A feature vector can be obtained for each image using the `model.features` function. The resulting size will vary depending on the architecture and the input image size. For some models there is a `model.features2` method that will extract features at a different point of the computation graph. Example UMAP visualizations [McInnes et al., 2018] of the features from different models is shown in Figure 1.

```
feats = model.features(img)
```

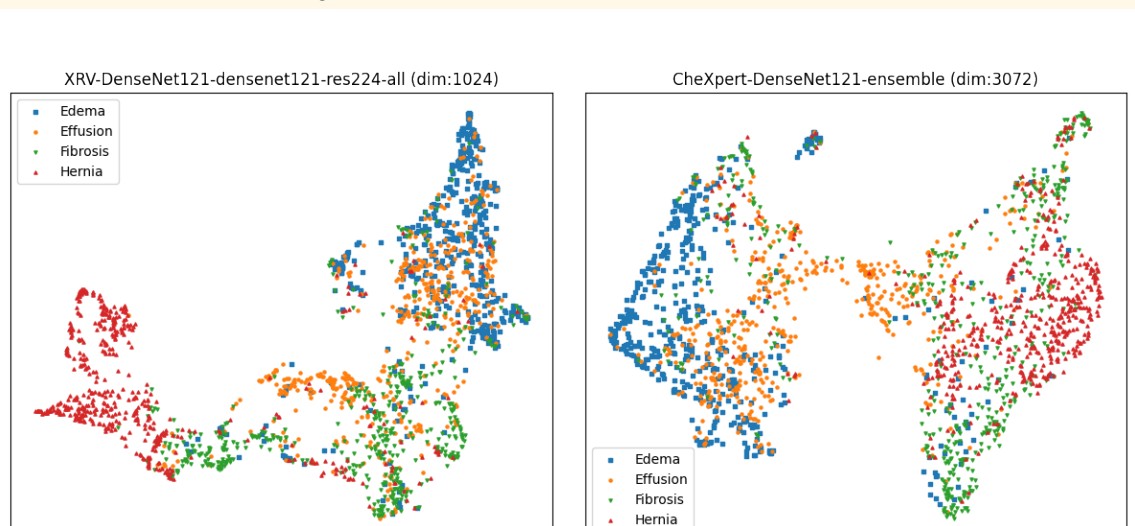

Figure 1: UMAP visualizations of the representations from different models. 2048 images, each containing only one of the 4 pathologies listed, are included in the UMAP.

## 2.5. Autoencoders

The library also provides a pre-trained autoencoder that is trained on the PadChest, NIH, CheXpert, and MIMIC datasets. This model was developed in [Cohen et al., 2021a] and provides a fixed latent representation. The goal of this model is to provide another representation extraction model which is not trained using supervised labels and has a decoder to reconstruct images from the latent representation.

```
ae = xrv.autoencoders.ResNetAE(weights="101-elastic")
z = ae.encode(image)
image2 = ae.decode(z)
```

## 3. Datasets

The datasets in this library aim to fit a simple interface where the `imgpath` and `csvpath` are specified. Some datasets require more than one metadata file and for some the metadata files are packaged in the library so only the imgpath needs to be specified.

Documentation for each dataset class provides citation information and download links. When possible, metadata is included with the library so only the `imgpath` needs to be specified. When possible based on the license, datasets have been uploaded to Academic Torrents [Cohen and Lo, 2014]. Otherwise, the dataset can be downloaded from its origin by following the links provided in the class documentation.

```
dataset = xrv.datasets.VinBrain_Dataset(imgpath="./train",
                                        csvpath="./train.csv")
```

Each dataset object also supports the standard `transform` argument which takes PyTorch transforms. While standard transforms will be applied there may be issues in how they deal with the specific tensor shapes returned by the dataloader (`[1,RES,RES]` where RES is the image resolution)

```
transform = torchvision.transforms.Compose([xrv.datasets.XRayCenterCrop(),
                                            xrv.datasets.XRayResizer(224)])
```

Tables 1 and 2 display the total count of images and labels available per dataset.

## 3.1. Image pre-processing

Both models and datasets expect the image pixel values to be between `[-1024,1024]`. The origin of this is arbitrary. The preprocessing of images used scales from the possible range of the images and not the min and max of the pixels. For example, if the image is 16-bit encoded then the possible pixels values are between 0 and 65,536 so this range will be mapped; not the min and max pixels of the specific image. The idea here is that we don't want to arbitrarily increase the contrast of an image because that could be removing information.

## 3.2. Dataset common fields

Each dataset contains a number of common fields. These fields are maintained when `xrv.datasets.SubsetDataset` and `xrv.datasets.MergeDataset` are used.

Table 1: Details of datasets that are included in this library. Number of images shows total images / usable frontal images. Useable frontal means images that are readable, have all necessary metadata, and are in AP, PA, AP Supine, or AP Erect view.

| Name | # Images (Total/Frontal) | | | Citation | Geographic Region |
|---|---|---|---|---|---|
| National Library of Medicine Tuberculosis | 800 | / | 800 | Jaeger et al. [2014] | USA+China |
| OpenI (National Library of Medicine) | 7,470 | / | 4,014 | Demner-Fushman et al. [2016] | USA |
| ChestX-ray8 (NIH) | 112,120 | / | 112,120 | Wang et al. [2017] | Northeast USA |
| RSNA Pneumonia Challenge | 26,684 | / | 26,684 | Shih et al. [2019] | Northeast USA |
| CheXpert (Stanford University) | 223,414 | / | 191,010 | Irvin et al. [2019] | Western USA |
| Google Labelling of NIH data | 4,376 | / | 4,376 | Majkowska et al. [2019] | Northeast USA |
| MIMIC-CXR (MIT) | 377,095 | / | 243,324 | Johnson et al. [2019] | Northeast USA |
| PadChest (University of Alicante) | 158,626 | / | 108,722 | Bustos et al. [2020] | Spain |
| SIIM-ACR Pneumothorax Challenge | 12,954 | / | 12,954 | Filice et al. [2020] | Northeast USA |
| COVID-19 Image Data Collection (CIDC) | 866 | / | 698 | Cohen et al. [2020c] | Earth |
| StonyBrook COVID-19 RALO Severity | 2,373 | / | 2,373 | Cohen et al. [2021b] | Northeast USA |
| Object-CXR (JF Healthcare) | 9,000 | / | 9,000 | - | China |
| VinBrain VinDr-CXR | 15,000 | / | 15,000 | Nguyen et al. [2020] | Vietnam |

- `dataset.pathologies` a list of strings identifying the pathologies contained in this dataset. This list corresponds to the columns of the `.labels` matrix. Although it is called pathologies, the contents do not have to be pathologies and may simply be attributes of the patient.

- `dataset.labels` field is a NumPy matrix [Harris et al., 2020] which contains a `1`, `0`, or `NaN` for each pathology. Each column is a pathology and each row corresponds to an item in the dataset. A `1` represents that the pathology is present, `0` represents the pathology is absent, and `NaN` represents no information.

- `dataset.csv` field which holds a Pandas DataFrame [McKinney, 2010] of the metadata `.csv` file that is included with the data. For some datasets multiple metadata files have been merged together. It is largely a "catch-all" for associated data and the referenced publication should explain each field. Each row aligns with the elements of the dataset so indexing using `.iloc` will work. Alignment between the DataFrame and the dataset items will be maintained when using tools from this library.

If possible, each dataset's `.csv` will have some common fields of the csv. These will be aligned when datasets are merged together.

- `dataset.csv.patientid` is a unique id that will uniquely identify patients in the dataset. This is useful when trying to prevent patient overlap between train and test sets or in conjunction with the next field to observe patients over time.

- `dataset.csv.offset_day_int` is an integer time offset for the image in the unit of days. This is expected to be for relative times and has no guarantee to be an absolute time although for some datasets it is and is formatted in unix epoch time.

- `dataset.csv.view` is a string indicating the projection/view in which the chest X-ray was acquired. Most will be "PA", "AP", or "AP Supine". A good discussion of views is contained in [Bustos et al., 2020].

### 3.3. Dataset tools

**Relabelling datasets** Working with dataset objects is a task that the library is designed to help with. Tasks such as aligning, composing, or taking a subset of a dataset are made easy using the functions discussed below. The function `xrv.datasets.relabel_dataset` will add, remove, and reorder the `.labels` field to have the same order as the pathologies argument passed to it. If a pathology is specified but doesn't exist in the dataset then a `NaN` will be put in place of the label.

```
# Note: dataset is directly changed, no return value
xrv.datasets.relabel_dataset(xrv.datasets.default_pathologies, dataset)
```

**Filtering based on views** Specific views can be specified in the constructor to select only those views. This is only supported on datasets which have view information. Common views have been standardized to "PA", "AP", or "AP Supine", but other non-standardized ones may exist. It is best to first load the dataset without filtering based on view and call `dataset.csv.view.unique()` to see what is available.

```
dataset = xrv.datasets.PC_Dataset(..., views=["PA","AP","AP Supine"])
```

**Ensuring one image per patient** The `unique_patients` argument will tell the dataset to only allow 1 image per patient. This only works on datasets which provide a patientid.

```
dataset = xrv.datasets.PC_Dataset(..., unique_patients=True)
```

**Obtaining summary statistics on a dataset** Simply printing the object will return counts for the available labels and their classes. This is also returned as a dictionary with the function `dataset.totals()`.

```
print(d_chex)
# Output:
CheX_Dataset num_samples=191010 views=['PA', 'AP']
{'Atelectasis': {0.0: 17621, 1.0: 29718},
 'Cardiomegaly': {0.0: 22645, 1.0: 23384},
 'Consolidation': {0.0: 30463, 1.0: 12982}, ...}
```

**Merging datasets together** The class `xrv.datasets.MergeDataset` can be used to merge multiple datasets together into a single dataset. This class takes in a list of dataset objects and assembles the datasets in order. This class will correctly maintain the `.labels`, `.csv`, and `.pathologies` fields and offer pretty printing.

```
dmerge = xrv.datasets.MergeDataset([dataset1, dataset2, ...])
# Output:
MergeDataset num_samples=261583
├0 PC_Dataset num_samples=94825 views=['PA', 'AP']
├1 RSNA_Pneumonia_Dataset num_samples=26684 views=['PA', 'AP']
├2 NIH_Dataset num_samples=112120 views=['PA', 'AP']
├3 SIIM_Pneumothorax_Dataset num_samples=12954
└4 VinBrain_Dataset num_samples=15000 views=['PA', 'AP']
```

**Taking a subset of a dataset**   When you only want a subset of a dataset the `SubsetDataset` class can be used. A list of indexes can be passed in and only those indexes will be present in the new dataset. This class will correctly maintain the `.labels`, `.csv`, and `.pathologies` fields and offer pretty printing.

```
dsubset = xrv.datasets.SubsetDataset(dataset, [0,5,60])
# Output:
SubsetDataset num_samples=3
└ of PC_Dataset num_samples=94825 views=['PA', 'AP']
```

For example this class can be used to create a dataset of only female patients by selecting that column of the csv file and using `np.where` to convert this vector into a list of indexes.

```
idxs = np.where(dataset.csv.PatientSex_DICOM=="F")[0]
dsubset = xrv.datasets.SubsetDataset(dataset, idxs)
# Output:
SubsetDataset num_samples=48308
└ of PC_Dataset num_samples=94825 views=['PA', 'AP']
```

## 3.4. Pathology and semantic masks

Masks for pathologies or semantic regions are also included for some datasets. These are useful for segmentation or for validating that a model is attributing importance to the correct area (as explored in [Viviano et al., 2020] and [Cohen et al., 2021a]).

Masks are not returned by the datasets by default; the constructor for the dataset must specify `pathology_masks=True` and/or `semantic_masks=True` for them to be returned. These are treated differently because pathology masks are associated with the `dataset.pathologies` while semantic masks are unrelated (such as segmentations of the lungs). If no pathology masks exist the data will not have those arguments available and won't be constructed. If images do not have masks available then the key `"pathology_masks"` on the sample will be empty. Example usage:

```
dataset = xrv.datasets.RSNA_Pneumonia_Dataset(imgpath="stage_2_train_images_jpg",
                                               views=["PA","AP"],
                                               pathology_masks=True)
```

Each sample will have a `pathology_masks` dictionary where the index of each pathology (in `dataset.pathologies`) will correspond to a mask of that pathology (if it exists). There may be more than one mask per sample, but only one per pathology. For simplicity, if there are multiple masks for a single pathology they will be merged together using a logical or. The resulting mask will be values between 0 and 1 of type `float`. Data augmentations will be performed to these masks as well using the same seed to ensure an identical transformation is applied.

```
sample["pathology_masks"][dataset.pathologies.index("Lung Opacity")]
```

## 4. Acknowledgements

We thank the following organizations for supporting this project in various ways: CIFAR (Canadian Institute for Advanced Research), Mila (Quebec AI Institute), University of Montreal, Stanford University's Center for Artificial Intelligence in Medicine & Imaging, and Carestream Health. We thank AcademicTorrents.com for making data available for our research.

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

## Appendix A. Extra Model Information

### A.1. Model Calibration

As described in [Cohen et al., 2019] and [Cohen et al., 2020b], Eq. 1 can be applied to calibrate the output of the model so that they can be compared with a piece-wise linear transformation. The goal is to make a prediction of 0.5 the estimated decision boundary based on held out test data. For each disease, we computed the optimal operating point of the ROC curve by maximizing the difference (*True positive rate - False positive rate*) on a hold out test set. It corresponds to the threshold which maximizes the informativeness of the classifier [Powers, 2011]. This is computed with the held out test subset of the dataset being used for training, as the model is intended to be evaluated on one of the other datasets. Also keep in mind that doesn't change the AUC on the test set, it is just for a nice calibrated probability output so you can use pred $> 0.5$ to get a prediction.

$$f_{opt}(x) = \begin{cases} \frac{x}{2opt} & x \leq opt \\ 1 - \frac{1-x}{2(1-opt)} & otherwise \end{cases} \tag{1}$$

# Appendix B. Extra Dataset Information

Table 2: Labels available for each dataset, the total number of positive examples for each indication across all datasets, and the total number of example in each dataset, and the sum over each row in the right column. The COVID-19 datasets are excluded from this table because they have many unique pathologies.

| | NIH | RSNA | NIH Google | PadChest | CheX | MIMIC | OpenI | NLMTB | SIIM | VinBrain | ObjectCXR | Total Positive Labels |
|---|---|---|---|---|---|---|---|---|---|---|---|---|
| Air Trapping | | | | X | | | | | | | | 3438 |
| Aortic Atheromatosis | | | | X | | | | | | | | 1728 |
| Aortic Elongation | | | | X | | | | | | | | 8116 |
| Aortic Enlargement | | | | | | | | | | X | | 3067 |
| Atelectasis | X | | | X | X | X | X | | | X | | 96,679 |
| Bronchiectasis | | | | X | | | | | | | | 1547 |
| Calcification | | | | | | | | | | X | | 452 |
| Calcified Granuloma | | | | | | | X | | | | | 193 |
| Cardiomegaly | X | | | X | X | X | X | | | X | | 86,196 |
| Consolidation | | | | X | X | X | | | | X | | 31,203 |
| Costophrenic Angle Blunting | | | | X | | | | | | | | 4244 |
| Edema | X | | | X | X | X | X | | | | | 82,689 |
| Effusion | X | | | X | X | X | X | | | X | | 156,156 |
| Emphysema | X | | | X | | | X | | | | | 3708 |
| Enlarged Cardiomediastinum | | | | | X | X | | | | | | 16,843 |
| Fibrosis | X | | | X | | | X | | | | | 2717 |
| Flattened Diaphragm | | | | X | | | | | | | | 535 |
| Foreign Object | | | | | | | | | | | X | 4500 |
| Fracture | | | X | X | X | X | X | | | | | 15,499 |
| Granuloma | | | | X | | | X | | | | | 2999 |
| Hemidiaphragm Elevation | | | | X | | | | | | | | 1609 |
| Hernia | X | | | X | | | X | | | | | 1881 |
| Hilar Enlargement | | | | X | | | | | | | | 4867 |
| ILD | | | | | | | | | | X | | 386 |
| Infiltration | X | | | X | | | X | | | X | | 34,296 |
| Lung Lesion | | | | | X | X | X | | | X | | 13,676 |
| Lung Opacity | | X | X | | X | X | X | | | X | | 158,919 |
| Mass | X | | | X | | | X | | | | | 6691 |
| Nodule/Mass | | | X | | | | | | | X | | 1431 |
| Nodule | X | | X | X | | | X | | | X | | 10,334 |
| Pleural Other | | | | | X | X | | | | | | 4586 |
| Pleural Thickening | X | | | X | | | X | | | X | | 8764 |
| Pneumonia | X | X | | X | X | X | X | | | | | 34,239 |
| Pneumothorax | X | | X | X | X | X | X | | X | X | | 38,513 |
| Pulmonary Fibrosis | | | | | | | | | | X | | 1617 |
| Scoliosis | | | | X | | | | | | | | 5569 |
| Tuberculosis | | | | X | | | | X | | | | 1165 |
| Tube | | | | X | | | | | | | | 6807 |
| **Total Examples** | 112,120 | 26,684 | 4376 | 108,722 | 191,010 | 243,324 | 4014 | 800 | 12,954 | 15,000 | 9000 | 728,004 |

Table 3: Counts of pathology and semantic masks available for each dataset.

| Task | NIH | RSNA | SIIM-ACR | VinBrain | CIDC |
|---|---|---|---|---|---|
| Pneumothorax | (BBox) 98 | | (Seg) 3,576 | (BBox) 96 | |
| Lung Opacity | | (Seg) 6,012 | | (BBox) 1,322 | |
| Atelectasis | (BBox) 180 | | | (BBox) 186 | |
| Effusion | (BBox) 153 | | | (BBox) 1,032 | |
| Cardiomegaly | (BBox) 146 | | | (BBox) 2,300 | |
| Infiltration | (BBox) 123 | | | (BBox) 613 | |
| Pneumonia | (BBox) 120 | | | | |
| Mass | (BBox) 85 | | | | |
| Nodule | (BBox) 79 | | | | |
| Nodule/Mass | | | | (BBox) 826 | |
| Aortic enlargement | | | | (BBox) 3,067 | |
| Pleural_Thickening | | | | (BBox) 1,981 | |
| Calcification | | | | (BBox) 452 | |
| Interstitial lung disease | | | | (BBox) 386 | |
| Consolidation | | | | (BBox) 353 | |
| Lung | | | | | (Seg) 425 |

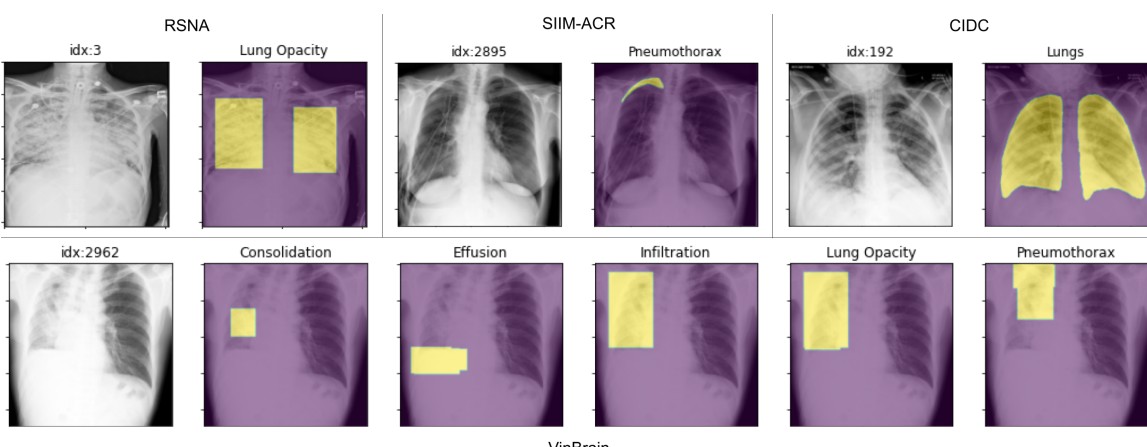

Figure 2: Example images and corresponding mask information available from multiple datasets. Some are bounding boxes and some are more exact segmentations.

## Appendix C. Distribution shift tools

A *covariate shift* between two data distributions arises when some extraneous variable confounds with the variables of interest in the first dataset differently than in the second [Moreno-Torres et al., 2012]. Covariate shifts between the training and test distribution in a machine learning setting can lead to models which generalize poorly, and this phenomenon is commonly observed in CXR models trained on a small dataset and deployed on another one [Zhao et al., 2019; DeGrave et al., 2020]. We provide tools to simulate covariate shifts in these datasets so researchers can evaluate the susceptibility of their models to these shifts, or explore mitigation strategies.

```
d = xrv.datasets.CovariateDataset(d1 = # dataset1 with a specific condition.
                                  d1_target = # target label to predict.
                                  d2 = # dataset2 with a specific condition.
                                  d2_target = #target label to predict.
                                  mode="train", # train, valid, or test.
                                  ratio=0.75)
```

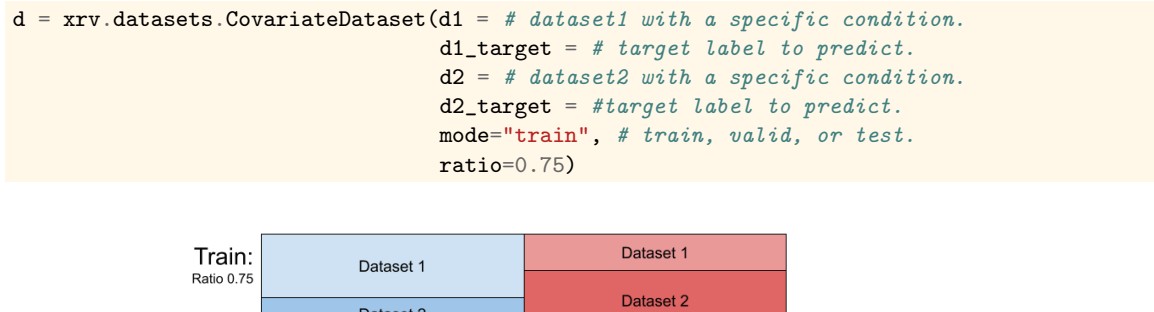

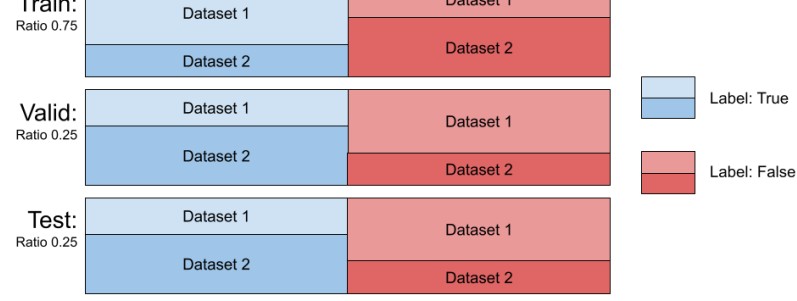

Figure 3: A CovariateDataset example of how the data would be split with a ratio of 0.75 specified. The target label will be balanced 50/50 in each split but the ratio of the origin dataset will be varied.

The class `xrv.datasets.CovariateDataset` takes two datasets and two arrays representing the labels. It returns samples for the output classes with a specified ratio of examples from each dataset, thereby introducing a correlation between any dataset-specific nuisance features and the output label. This simulates a covariate shift. The test split can be set up with a different ratio than the training split; this setup has been shown to both decrease generalization performance and exacerbate incorrect feature attribution [Viviano et al., 2020]. See Figure 4 for a visualization of the effect the ratio parameter has on the mean class difference when correlating the view (each dataset) with the target label. The effect seen with low ratios is due to the majority of the positive labels being drawn from the first dataset, where in the high ratios, the majority of the positive labels are drawn from the second dataset. With any ratio, the number of samples returned will be the same in order to provide controlled experiments. The dataset has 3 modes, train sampled using the provided ratio and the valid and test dataset are sampled using 1−ratio.

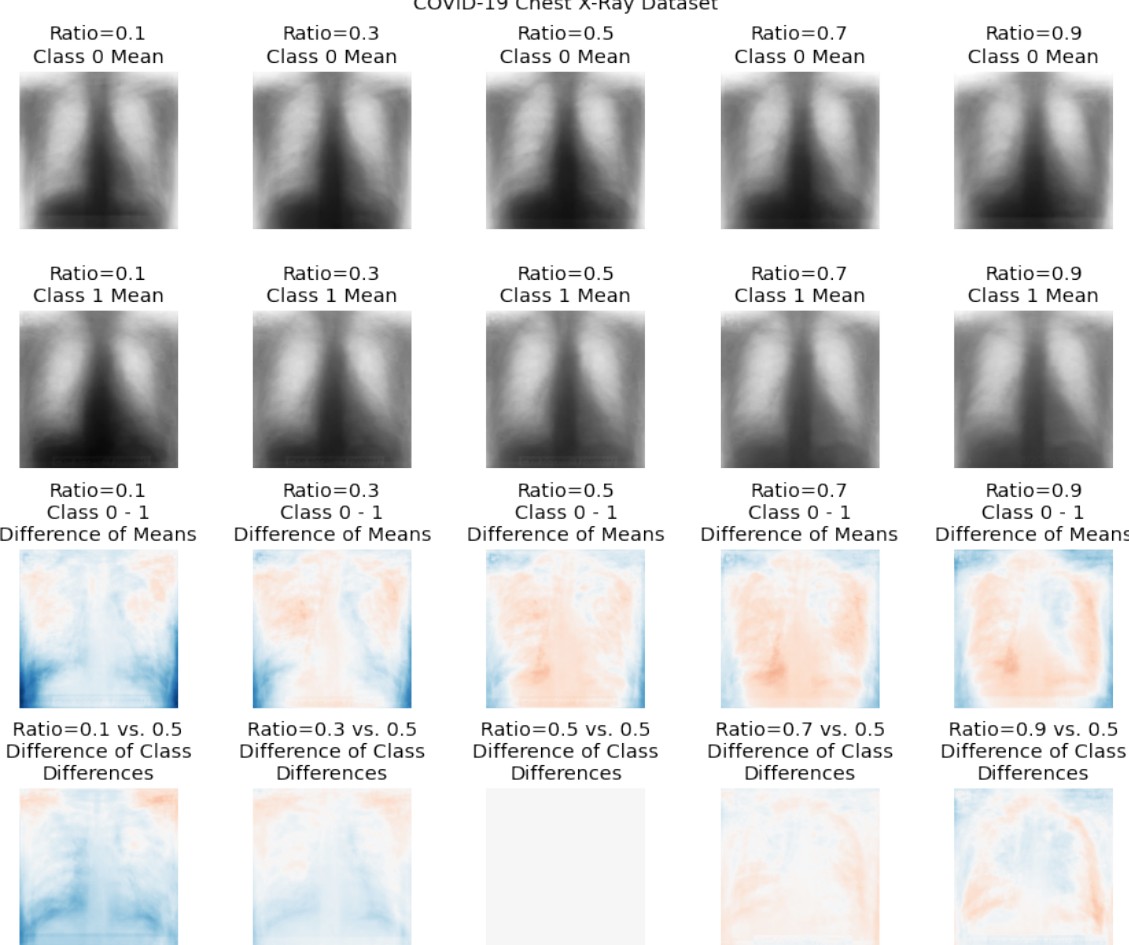

Figure 4: An example of the mean class difference drawn from the COVID-19 dataset at different covariate ratios. Here, the first COVID-19 dataset consisted of only AP images, whereas the second dataset consisted of only PA images. The third row shows, for each ratio, the difference in the class means, demonstrating the effect of sampling images from the two views on the perceived class difference. The fourth row shows the difference between each ratio's difference image, and the difference image with a ratio of 0.5 (balanced sampling from all views).

## Appendix D. Transfer Learning Example

This is an example of fine-tuning a pre-trained model on a dataset which is in the TorchXRayVision format. This code is available online[2].

```python
# Use XRV transforms to crop and resize the images
transforms = torchvision.transforms.Compose([xrv.datasets.XRayCenterCrop(),
                                             xrv.datasets.XRayResizer(224)])

# Load Google dataset and PyTorch dataloader
dataset = xrv.datasets.NIH_Google_Dataset(imgpath=dataset_dir,
                                          transform=transforms)
dataloader = torch.utils.data.DataLoader(dataset, batch_size=8)

# Load pre-trained model and erase classifier
model = xrv.models.DenseNet(weights="densenet121-res224-all")
model.op_threshs = None # prevent pre-trained model calibration
model.classifier = torch.nn.Linear(1024,1) # reinitialize classifier

optimizer = torch.optim.Adam(model.classifier.parameters()) # only train classifier
criterion = torch.nn.BCEWithLogitsLoss()

# training loop
for batch in dataloader:
    outputs = model(batch["img"])
    targets = batch["lab"][:, dataset.pathologies.index("Lung Opacity"), None]
    loss = criterion(outputs, targets)
    loss.backward()
    optimizer.step()
```

---

2. https://github.com/mlmed/torchxrayvision/blob/master/scripts/transfer_learning.ipynb

