# OpenReview forum: "TorchXRayVision: A library of chest X-ray datasets and models"
_MIDL.io/2022/Conference — MIDL 2022_

### Official Review · Reviewer_86hC · 2022-01-21

**Confidence:** 3
**Preliminary Rating:** 4
**Recommendation:** Poster

**Summary:**

This paper describes an open source library that provides a unified API for working with a number of chest X-ray datasets. The library also offers a number of pretrained models. The value of the library lies in the abstraction between different datasets with different image and metadata paths and formats, different pathologies, and different annotations. The paper mostly describes the library's functionality and motivates it with a few use cases and examples.  It is evident from the paper and the library that the authors are very familiar with chest X-ray datasets and applications.

**Strengths:**

The authors make it easier for others to work with more than one chest X-ray dataset. This facilitates evaluating models on data from multiple sources and developing more robust models that were trained on more than one dataset. The pretrained models may be useful as baseline models to compare against.

**Weaknesses:**

I want to be very open about this: I am really unsure how to grade this paper and whether it is interesting to the MIDL audience. I have experienced myself how hard it is to get papers accepted that just describe an implementation, and I was always of the opinion that this should be easier. The authors definitely provide a value to the community through their library, which I want to acknowledge, but I must admit that I was not fascinated reading the paper. I think I would have liked it better if it contained less "how to" information (which belongs into the software documentation) and a little more focus on the motivation and concepts (which much of the paper already does well

A weakness of the library might be that some parts of it seem to be geared towards pytorch, although the core dataset handling and abstraction mechanisms could also have been useful to tensorflow users.  Along similar lines, I would suggest to have a low-level API that provides just filenames to be loaded, so that the library could also be integrated into frameworks where native image loaders (e.g., ITK ones) should be used and numpy arrays are not desired.

**Deanonymize Review:**

yes

**Detailed Comments:**

The 3.1 Image Preprocessing section reads a little strange, but that might be because operation it describes may be unexpected. Will 0 of uint16 input really be mapped to -1024?  Is this not configurable?  I assume the mapping is linear and does not do any clamping (because no out of range values can exist), but at least the former should be explicitly stated.

I like the idea of mapping pathology labels and including NaN to mark non-existing outputs.  (If you don't know it yet, check the numpy.ma module BTW.)

The unique_patients option is also useful (although it does not seem to allow specifying *which* images to drop / retain).

**Final Rating After The Rebuttal:**

4: Weak Accept

**Justification Of The Final Rating:**

I see that some points of the reviewers were addressed, and while I am still a little bit unsure about the value of this paper for the MIDL audience, because it is more similar to a software documentation than to a scientific publication, I stay with my rating of "weak accept". The software itself is relevant to the community and the author's effort in sharing it deserves support.

**Paper Type:**

methodological development

**Questions To Address In The Rebuttal:**

Maybe an architecture diagram with components / software layers would also have been useful.  I could sacrifice many of the code examples for some more conceptual description.

It would be nice if the library did not have so many hard dependencies.  For instance, it is not possible to import it without having pydicom installed, even though I believe many datasets would work without pydicom.  Similar for torchvision, see above.

I think all preprocessing should be optional and configurable, including the value mapping.

**Special Issue:**

no

---

### Official Review · Reviewer_q8AK · 2022-01-27

**Confidence:** 4
**Preliminary Rating:** 5
**Recommendation:** Poster

**Summary:**

This work presents **TorchXRayVision** - a PyTorch based library with the aim of providing a singular interface for working with the majority of the Chest X-Ray imaging datasets available today.

* As its central contribution, the library provides within a singular API access to data loaders for 13 public Chest X-Ray datasets with heterogeneous structures, labels and formats. Several commonly-used functionalities such as data filters, merging, subsets are provided along with standard pre-trained models as baselines within the library.

* The paper presents examples, code snippets and visualizations to explain functionalities and benefits to multi-domain, transfer-learning applications.

* The novelty of such a work is generally arguable. However in this case, the relative absence of an equivalent library for Chest X-Ray imaging, the benefits to standardization, convenience of implementation and expediting research in Chest X-Ray Imaging Analysis argue in favor of this work’s significance.


**Strengths:**

#### __Primary Strength__
The primary strength of the paper is the creation of a single api to access Chest X-ray datasets, along with relevant baselines, pre-trained models and standard data tools that make the research/development pipeline on Chest X-Ray imaging convenient, standardized and easily reproducible.

#### __Significance of Work__
Preparing data classes to adapt to heterogeneous structures, formats, labels, and source-specific fields are a major component of working with Chest X-Ray datasets. The difficulty is compounded when working with multiple datasets, as is the case in applications such as domain adaptation, continual learning, and transfer learning. A large part of this effort is mitigated by TorchXRayVision. Standardization of data handling, preprocessing, corresponding standard baselines and pretrained models provided in this library will aid reproducibility.
Another argument in support of this paper’s strength, is the benefit the community has already seen from this library [Sections 1.2, 1.3]. Papers across Transfer Learning [1], patient clinical trajectory [2, 4], few-shot transfer [3], out of distribution generalization [5], continual learning [6], etc have already leveraged this library.

*To my knowledge, there is no equivalent library for Chest X-Ray imaging, and the benefits to reproducibility, standardization, convenience of implementation and expediting research are significant.*

#### __Structure/Language of Paper__
The paper itself is easy to follow, and summarizes the primary functionalities of the library well. Example applications with sample code are provided in the paper as reference to each functionality, and a few example notebooks are provided within the library. The library is integrated well with standard PyTorch pipelines and follows conventional pythonic styles.

#### __References__
[1] Douglas P. S. Gomes et al. MAVIDH Score: A COVID-19 Severity Scoring using Chest X-Ray Pathology Features.

[2] Douglas P. S. Gomes, et al.  Potential Features of ICU Admission in X-ray Images of COVID-19 Patients

[3] Mehdi Cherti and Jenia Jitsev. Effect of large-scale pre-training on full and few-shot transfer learning for natural and medical images. arXiv:2106.00116, 5 2021.

[4] Aniket Maurya. Predicting intubation support requirement of patients using Chest X-ray with Deep Representation Learning.

[5] Joseph Paul Cohen, Mohammad Hashir, Rupert Brooks, and Hadrien Bertrand. On the limits of cross-domain generalization in automated X-ray prediction. Medical Imaging with Deep Learning, 2020

[6] Srivastava, S., Yaqub, M., Nandakumar, K., Ge, Z., & Mahapatra, D. (2021). Continual domain incremental learning for chest x-ray classification in low-resource clinical settings. FAIR workshop MICCAI, 2021


**Weaknesses:**

#### __Novelty__

The novelty of this work, as expected, is arguable. The library provides no 'novel’ functionality: The implementations of the data classes, tools and pretrained models are largely a development effort (modeled on torchvision [1]), and in most cases a re-implementation of standard protocols for Chest X-Ray datasets ([2],[3], among other standard dataset structuring, preprocessing, reading protocols), albeit within a single structure and API.

#### __Documentation__

The documentation within the library itself is lacking -
* To start to use the library seems to require a detailed read of the dataset module (https://github.com/mlmed/torchxrayvision/blob/master/torchxrayvision/datasets.py). To my knowledge, the structure the data is expected in, is confirmed only within the data classes. This seems unnecessary. However, a link to the class documentation or a more detailed description within the repository ReadMe may be helpful.
* The core dataset module might benefit from documentation on the Merge, Filter, Subset classes.
* The Benchmarks and Baselines implemented in the library could benefit from more detailed documentation. The benchmarks.md for example lacks training details, graphs/logs, employed metrics, multi-seed results - details that will be needed in using the provided pretrained models as reportable benchmarks.

#### __References__

[1] https://pytorch.org/vision/stable/index.html

[2] Irvin, Jeremy, et al. "Chexpert: A large chest radiograph dataset with uncertainty labels and expert comparison." Proceedings of the AAAI conference on artificial intelligence. Vol. 33. No. 01. 2019.

[3] Bustos, Aurelia, et al. "Padchest: A large chest x-ray image dataset with multi-label annotated reports." Medical image analysis 66 (2020): 101797.


**Deanonymize Review:**

yes

**Detailed Comments:**

--

**Final Rating After The Rebuttal:**

5: Strong Accept

**Justification Of The Final Rating:**

I thank the authors for their response and once again for their work. I'm going to maintain my previous rating given the value of this library to the general Chest X-ray community. Finally, given that the documentation is still understandably at the same state, I'm hopeful of it being improved in the future.

**Paper Type:**

validation/application paper

**Questions To Address In The Rebuttal:**

Would be helpful to address the documentation aspect primarily.

If the library is to be leveraged as a standard for reproducibility and benchmarking, the documentations for model benchmarking and code use should ideally be more thorough (within the repository particularly).

#### __Some Housekeeping:__

Sparsely occurring typos/formatting issues through the paper.
A few I could find:
- **Section 2.1:** `... and scaled larger OR smaller…’
- **Table 1:** Formatting of `Citation’ column.


**Special Issue:**

no

---

### Meta-Review · Area_Chair_sNv6 · 2022-02-16

**Recommendation:** Accept (Oral)
**Confidence:** 4

**Metareview:**

The paper is a good contribution to the medical imaging community since it describes in detail about the use of the TorchXRayVision library. Although not strictly a paper contributing "novel methods", this is definitely a good piece of work in a very applied field such as medical image analysis. With the exception of one reviewer, all others rate the paper very highly and I feel  the paper should be accepted and the audience given an opportunity to know about it.

---

### Decision · Program_Chairs · 2022-02-28

Accept